# ZERO-LABEL PROMPT SELECTION

## ABSTRACT

Natural language prompts have been shown to facilitate cross-task generalization for large language models. However, with no or limited labeled examples, the cross-task performance is highly sensitive to the choice of prompts, while selecting a high-performing prompt is challenging given the scarcity of labels. To address the issue, we propose a Zero-Label Prompt Selection (ZPS) method that selects prompts without any labeled data or gradient update. Specifically, given the candidate human-written prompts for a task, ZPS labels a set of unlabeled data with a prompt ensemble and uses the pseudo-labels for prompt selection. Experiments show that ZPS improves over prior methods by a sizeable margin in zero-label performance. We also extend ZPS to a few-shot setting and show its advantages over strong baselines such as prompt tuning and model tuning.

## 1 INTRODUCTION

Recently, extensive studies have shown that large language models (LLMs) have promising performance for few-shot learning (Brown et al., 2020; Zhao et al., 2021; Schick & Schütze, 2021; Gao et al., 2021), and they even show strong generalization abilities to new tasks without any annotated data (Brown et al., 2020; Wei et al., 2021; Sanh et al., 2021). Different from conventional fine-tuning methods that require expensive parameter updates for each downstream task, *prompts* are employed to provide in-context information or task instructions, which is helpful for guiding models to perform each task. Manually-written prompts are often used to specify the task and unify the format of inputs.

However, the performance of different prompts during evaluation can vary from near state-of-the-art to random guess; e.g., using a non-optimal prompt can cause a performance drop of up to 60 points on the CB task (Zhao et al., 2021). Previous work mainly relies on using multiple prompts (Brown et al., 2020; Wei et al., 2021; Sanh et al., 2021) or a prompt ensemble (Zhou et al., 2022) to enhance the performance and robustness when generalizing to test tasks, while omitting the fact that using multiple prompts leads to a substantially increased computational cost, which hinders the practical deployment of LLMs. These challenges make prompt selection an important problem.

There have been efforts on improving model performance via searching for a better prompt. For example, Jiang et al. (2020) proposed two automatic methods to augment prompts. They further explored combining the generated diverse prompts with ensemble methods. Shin et al. (2020) designed a gradient-based search method to find trigger words in a prompt. Gao et al. (2021) developed a way to use a span-corruption pretraining objective for prompt generation. Deng et al. (2022) presented RLprompt, a prompt search method with reinforcement learning which relies on a policy network trained with a carefully designed reward function. Prasad et al. (2022) designed an iterative prompt search algorithm that relies on human-defined edit rules to improve the few-shot performance. Xu et al. (2022) proposed GPS, a genetic prompt searching algorithm that leveraged generative language models for prompt augmentation. Nevertheless, the main drawback of such methods is that they all require an additional labeled set to serve as a prompt scoring set or to provide the rewards or gradient signals. It remains challenging when no labeled samples are available. Thus, a crucial question arises:

*Is it possible to select a high-performing prompt without any labeled data or gradient update?*

In this paper, we answer this question affirmatively. To tackle the aforementioned problem, we propose ZPS—Zero Label Prompt Selection—a simple-yet-effective technique for selecting a high-

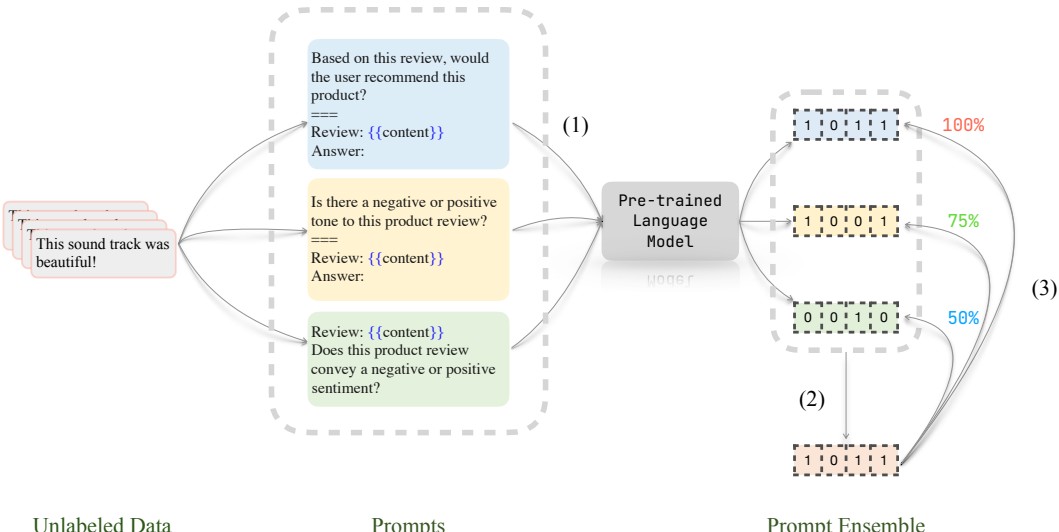

Figure 1: The main pipeline of ZPS. **(1)** A number of prompted unlabeled data are fed into the pretrained language model to get logits and predictions. **(2)** The pseudo labels are obtained from the prompt ensemble. **(3)** The pseudo labels are used to calculate the pseudo accuracy of prompts. And the one with the highest pseudo accuracy is selected. Some details like prompt filtering are omitted for brevity and can be found in the text.

performing prompt in a zero-label setting. As illustrated in Figure 1, given a set of candidate human-written prompts $\mathcal{P}$ and an unlabeled dataset $\mathcal{X}$ for a task, the ensemble of prompts is used to annotate the unlabeled data. Finally, the pseudo-labels generated from the prompt ensemble are used for prompt selection. We also extend the idea of ZPS to a few-shot setting and validate our advantages over strong baselines such as prompt tuning and model tuning. In the few-shot setting, we further explore the role of pseudo-labeled data: pseudo-labeled data can not only be used for prompt selection, but checkpoint selection as well. By using pseudo-labeled data for checkpoint selection, there is no need to split a subset from limited labeled data as a validation set, which means more labeled data can be used for training to boost model performance.

Our contributions are summarized as follows.

- We propose a novel Zero-Label Prompt Selection (ZPS) algorithm to select high-performing prompts without an extra validation set or any parameter update.

- We show that ZPS can be used in a plug-and-play manner to boost zero-label and few-shot performance. Extensive experiments show that ZPS leads to a substantial performance boost for both the zero-label and few-shot settings.

## 2    RELATED WORK

**Pseudo-labeling.**    Recently, there have been many advances in deep learning with pseudo-labeling. Pseudo-labeling (Lee et al., 2013; Reed et al., 2015; Shi et al., 2018) employs a model to make predictions for unlabeled samples (Yarowsky, 1995b; McClosky et al., 2006). Iscen et al. (2019) showed that pseudo-labels can also be created by label propagation instead of direct network predictions. Shi et al. (2018) incorporate the idea of confidence levels for unlabeled samples to discount influences from uncertain samples. Another line of work is self-training (III, 1965; Yarowsky, 1995a; Riloff, 1996), which trains a model with both labeled and pseudo-labeled data for a few iterations. Some modifications like strong data augmentation (Zoph et al., 2020), learning an equal-or-larger student model (Xie et al., 2020), and using additional noise (Xie et al., 2020; He et al., 2020) are shown to be beneficial for self-training. Another popular technique is ensemble distillation (Hinton et al., 2015), which means distilling knowledge in an ensemble into a single model.

**Zero-label Learning.** Different from pseudo-labeling where labeled data is usually available, zero-label learning transfers a pretrained model to unseen tasks with unlabeled data only. Instead of directly predicting labels for unlabeled data, Wang et al. (2021) proposed to use unlabeled data to generate synthetic data without tuning the generative models. Zhou et al. (2022) designed a prompt-consistency loss for finetuning on unlabeled data. Lang et al. (2022) used co-training to fine-tune both a large language model and a smaller task-specific model, where the different views of the data are obtained from different prompts. These methods study generating synthetic data or improved training techniques, while our ZPS focuses on an orthogonal aspect—how to select high-performing prompts.

**Prompt Search.** Past works attempt to improve prompt quality via *prompt tuning* (Liu et al., 2021b; Lester et al., 2021; Li & Liang, 2021; Vu et al., 2021; Gu et al., 2021; Mokady et al., 2021; Qian et al., 2022; An et al., 2022). Soft-prompts are optimized via gradient descent. However, the continuous nature of soft-prompts makes them hard to be interpreted. Other prior work has looked into optimizing discrete manual prompts in various aspects, such as selecting priming examples (Zhao et al., 2021; Liu et al., 2021a), ordering examples (Zhao et al., 2021; Lu et al., 2022; Kumar & Talukdar, 2021), and prompt search (Jiang et al., 2020; Gao et al., 2021; Shin et al., 2020; Prasad et al., 2022; Xu et al., 2022). The aforementioned methods require updating parameters and a validation set. Our ZPS, on the contrary, is a tuning-free method while no labeled data is required.

## 3 METHOD

In this section, we provide a detailed description of the presented approach, Zero-Label Prompt Selection (ZPS). ZPS mainly consists of three parts: prompt filtering, ensemble, and selection. After filtering out some low-performing prompts, the ensemble of the remaining prompts is used for prompt selection. We first start with our problem definition of zero-label learning.

### 3.1 ZERO-LABEL LEARNING

Suppose an LLM has been pretrained. Given a new task and a set of unlabeled data $\mathcal{X}$ associated with the task, we use the LLM to perform the new task without any label. Zero-label learning is a reasonable setting since task-relevant unlabeled data are usually accessible at test time. This setting has also been adopted in previous work (Meng et al., 2022) to test the ability of cross-task generalization.

### 3.2 PROMPT FILTERING

In this part, we introduce prompt filtering. We assume that we are given the pretrained model $\mathcal{M}$, a set of prompts $\mathcal{P}_i = (\mathcal{T}_i, \mathcal{V}_i) \in \mathcal{P}$, and a set of unlabeled data $\mathcal{X} = \{x_1, \ldots, x_n\}$ with size $n$. Formally, a prompt $\mathcal{P} = (\mathcal{T}, \mathcal{V})$ is a pair composed of two parts: a prompt template and the corresponding verbalizer. The prompt template is defined as a function that transforms the unlabeled data into a sequence. For example, given a prompt template $\mathcal{T}$ of a sentiment classification task "*Based on this review, would the user recommend this product? Review:*{{*content*}} *Answer:*", we will fill a predicted text sample into the "{{content}}" blank. Further, the verbalizer $\mathcal{V}$ is defined as a function that maps each label to a phrase. For instance, the verbalizer corresponding to $\mathcal{T}$ will map 1 to "*Yes*", and 0 to "*No*". As a result, the LLM performs the tasks by predicting the mapped phrases.

Now we want to filter low-performing prompts before we proceed to the next step. Given an unlabeled sample $x_k$, the LLM outputs the conditional probability for a given label $y_j$:

$$P(y_j \,|\, x_k, \mathcal{P}_i) = \mathbb{P}_{\mathcal{M}}(\mathcal{V}_i(y_j) \,|\, \mathcal{T}_i(x_k)),$$

where we use $\mathbb{P}_{\mathcal{M}}(\cdot \,|\, \cdot)$ to denote the LLM probability. We define the confidence score of a prompt as

$$c_i = \sum_{x_k \in \mathcal{X}} \left( p(y_{(1)} \,|\, x_k, \mathcal{P}_i) - p(y_{(2)} \,|\, x_k, \mathcal{P}_i) \right),$$

where $y_{(1)}$ and $y_{(2)}$ denote the choices with the highest and the second highest LM probability, separately. The confidence scores represent how confident the LLM is about the predictions with a

specific prompt. Let $(c_1, \cdots, c_p)$ denote the confidence scores of all $p$ prompts. We use k-means to cluster these scores into two sets and discard the set of prompts with lower scores.

### 3.3 PROMPT ENSEMBLE AND SELECTION

In this subsection, after clustering and filtering, we will introduce how to use the rest of the prompts to form a prompt ensemble, and utilize the prompt ensemble for prompt selection. We then provide three possible ways to combine prompts as a prompt ensemble. We note that directly serving a prompt ensemble at inference time will increase the computational cost or latency substantially, which is not practical for many applications. In this work, we mainly focus on the setting where only one prompt can be served.

The key idea of our approach is from one of most classic techniques—ensemble (Hansen & Salamon, 1990; Krogh & Vedelsby, 1994). We ensemble all available prompts to obtain a pseudo label for each sample. Since we cannot directly calculate accuracy on a labeled validation set, we obtain a "pseudo accuracy" by calculating the agreement between model predictions and pseudo labels, as illustrated in Figure 1. Specifically, the prediction of an input $x_k$ given the prompt $\mathcal{P}_i$ is

$$\hat{y}_{i,k} = \underset{y \in \mathcal{Y}}{\operatorname{argmax}} \, P(y \,|\, x_k, \mathcal{P}_i) \,.$$

We use the notation $\mathbf{y}_i = (\hat{y}_{i1}, \ldots, \hat{y}_{in})$ to denote all the predictions of prompt $\mathcal{P}_i$, and we use $\mathbf{y}_e$ to denote the prediction from the prompt ensemble. The "pseudo accuracy" of the prompt $\mathcal{P}_i$ is computed as

$$\operatorname{acc}'_i = f_{\operatorname{acc}}(\mathbf{y}_i, \mathbf{y}_e) \,.$$

where we use $f_{\operatorname{acc}}$ to denote the accuracy function that calculates the proportion of identical predictions between the two vectors. Then we select the prompt with the highest "pseudo accuracy" as the prompt we use for the target task. Intuitively, the pseudo labels are assumed to have higher accuracy compared to the prediction from a single prompt, which provides a stronger signal for selection.

There are many possible ways to form a prompt ensemble for obtaining $\mathbf{y}_e$. Here we discuss three possible ensemble strategies.

**Log-probability mean.** Given an unlabeled example $x_k$, we use the average of log probabilities to calculate the score of a choice $y$, and obtain the ensemble prediction $\hat{y}_{e,k}$:

$$s(x_k, y) = \frac{1}{p} \sum_{\mathcal{P}_i \in \mathcal{P}} \log P(y \,|\, x_k, \mathcal{P}_i) \,, \qquad \hat{y}_{e,k} = \underset{y \in \mathcal{Y}}{\operatorname{argmax}} \, s(x_k, y) \,.$$

**Probability mean.** It is also possible to get the score by averaging the LLM probabilities from different prompts. Specifically, we have

$$s(x_k, y) = \frac{1}{p} \sum_{\mathcal{P}_i \in \mathcal{P}} P(y \,|\, x_k, \mathcal{P}_i) \,.$$

**Majority vote.** This method aggregates the prediction of each prompt given an input $x_k$ and outputs the choice with the maximum occurrences. Formally,

$$s(x_k, y) = \sum_{\mathcal{P}_i \in \mathcal{P}} \mathbb{1}(\hat{y}_{i,k} = y) \,,$$

where $\mathbb{1}(\cdot)$ is the indicator function and $\hat{y}_{i,k}$ denotes the prediction of prompt $\mathcal{P}_i$ given the input $x_k$.

### 3.4 ZPS FOR FEW-SHOT LEARNING

In addition to zero-label learning, we extend ZPS to the few-shot learning setting. In this setting, we are given a labeled set with $m$ samples, say $\mathcal{D}_m$, in addition to the unlabeled data $\mathcal{X}$ and the LLM $\mathcal{M}$. Our goal is to develop a strategy to boost the performance of the model. Traditional model tuning often splits $\mathcal{D}_m$ into to parts of the same size, $\mathcal{D}_{m/2}$ and $\mathcal{D}'_{m/2}$, as a training set and a validation set. The model is first trained on $\mathcal{D}_{m/2}$, and the validation set $\mathcal{D}'_{m/2}$ is then used for checkpoint selection

and prompt selection. However, since the number of labeled examples is limited, using a validation set means less data is used for training. With ZPS, we propose to use pseudo-labeled data for both checkpoint and prompt selection. As a result, all labeled examples are used for tuning the models, which leads to better generalization performance.

# 4    EXPERIMENTS

In this section, we conduct extensive quantitative experiments to evaluate our ZPS against baseline methods. We mainly consider two experimental settings, zero-label learning and few-shot learning, as described in Section 3. In addition, we also investigate and analyze several influential factors and hyper-parameters in ZPS.

## 4.1    EXPERIMENTAL SETUP

### 4.1.1    DATASETS

We use the T0 benchmark (Sanh et al., 2021) that consists of 39 training tasks of 8 task types, and 11 test tasks of 4 task types. Both the training and the test sets are disjoint in task types. Specifically, the test tasks include natural language inference (RTE (Dagan et al., 2006), CB (De Marneffe et al., 2019), ANLI/R1-R3 (Nie et al., 2020)), coreference resolution (WSC (Levesque et al., 2012), Winogrande (Sakaguchi et al., 2020)), sentence completion (COPA (Roemmele et al., 2011), StoryCloze (Mostafazadeh et al., 2017), Hellaswag (Zellers et al., 2019)), and word sense disambiguation (WiC (Pilehvar & Camacho-Collados, 2019)). We also use the prompt candidates provided in T0, which are constructed using PromptSource (Bach et al., 2022). For the zero-label setting, we construct the unlabeled data by removing labels of the training-split data for each test task. For the few-shot setting, we randomly sample 32 labeled data from the training split.

### 4.1.2    BASELINES

For the zero-label setting, we compare our ZPS with the following baseline methods.

• **T0**  T0 (Sanh et al., 2021) employs prompted multi-task pretraining for cross-task generalization, which provides a framework for all methods considered in our experiments. T0 does not provide a way to select prompts without labels, and the average performance of all candidate prompts is used.

• **Self-Training**  Self-training first uses a trained T0 to label the unlabeled data, and then uses the pseudo-labeled data to further finetune the T0 model (Sanh et al., 2021). The whole process is repeated multiple times. The self-training method also reports average performance over multiple prompts.

• **Ensemble Distillation**  We use a prompt ensemble (Hinton et al., 2015) to pseudo label the data and distill the knowledge into multiple prompts using prompt tuning (Lester et al., 2021). The tuned soft prompt embeddings are concatenated to different prompts and the mean performance of all prompts is reported.

For the few-shot setting, we mainly compare ZPS with the following two categories of baseline methods, respectively methods with parameter tuning, including model tuning and prompt tuning, and methods without parameter tuning, including in-context learning, GRIPS, and GPS.

• **In-Context Learning (ICL)**  ICL (Brown et al., 2020) is a few-shot method which requires a few input-output pairs as priming demonstrations to guide models to perform the test task. Given a prompt $\mathcal{P}$ and a test sample, a few prompted labeled samples are concatenated to the prompted test sample to form an input. The average performance of all prompts is reported.

• **GRIPS**  GRIPS (Prasad et al., 2022) is an edit-based prompt search approach. It iteratively mutates the prompts with human-defined edit rules. Then, the augmented prompts with the highest scores on the validation set are selected. It also reports the average performance on multiple augmented prompts.

• **GPS**  GPS (Xu et al., 2022) is also a gradient-free prompt search method that adapts the genetic algorithm and uses generative models to augment prompts. We report the average performance of the selected prompts.

• **Model Tuning (MT)** Model tuning finetunes all parameters of a language model for each task. We follow the few-shot setting in Zheng et al. (2021) to use half of the data for training and the other half for model selection.

• **Prompt Tuning (PT)** Prompt tuning (Liu et al., 2021b; Lester et al., 2021) tunes a continuous embedding while the LLM is frozen. Training and validation splits are identical to model tuning.

### 4.1.3 TRAINING DETAILS

For fair comparison, we keep the number of labeled samples as 32 for all few-shot methods. For all methods that require gradient update (PT, MT, self-training), we use the Adafactor Optimizer and set the batch size as 4. We set the learning rate as 5e-5 for MT and self-training, and 0.05 for PT. For PT, we set the number of soft prompt tokens as 3 for zero-label ensemble distillation and 1 for few-shot learning. For ICL, we randomly select 2 examples from the training set of each task to compose the priming prompt. The above hyperparameters are chosen based on validation results. For GRIPS and GPS, we also follow the hyper-parameters reported in Prasad et al. (2022) and Xu et al. (2022).

### 4.2 MAIN RESULTS AND ANALYSIS

**Zero-Label Performance.** In this section, we compare ZPS with the baselines mentioned in the last subsection on the 11 test tasks. As shown in Table 1, our ZPS outperforms the T0 baseline, which shows the effectiveness of prompt selection. Our zero-label ZPS even has a considerable advantage over some few-shot methods (GRIPS, ICL, PT) in Table 2, which shows that the selection of prompts is a crucial factor in model performance. Moreover, combining ZPS with ensemble distillation further boosts performance and outperforms using ensemble distillation alone, which indicates that ZPS is able to select high-performing prompts in a prompt tuning scenario.

| Method | Natural Language Inference | | | | | Sentence Completion | | | Co-reference | | WSD | Avg. |
|---|---|---|---|---|---|---|---|---|---|---|---|---|
| | RTE | CB | ANLI1 | ANLI2 | ANLI3 | COPA | Hella. | Story. | WSC | Wino. | WiC | |
| Self-training | 84.12 | **85.71** | 42.41 | 39.35 | 43.04 | 84.01 | **50.55** | 97.49 | 49.04 | 55.20 | 55.49 | 62.40 |
| T0 | 80.97 | 70.12 | 43.16 | 38.68 | 41.87 | 90.02 | 33.55 | 92.84 | 61.06 | 59.70 | 56.13 | 60.74 |
| EnsD | 83.86 | 75.48 | 42.22 | 38.78 | 41.67 | 92.11 | 39.71 | 95.65 | 58.65 | 59.18 | 59.69 | 62.45 |
| T0 + ZPS (Ours) | **86.28** | 80.36 | **45.10** | **40.50** | 42.92 | 92.00 | 34.33 | 95.83 | **62.50** | 59.75 | 59.40 | 63.54 |
| EnsD + ZPS (Ours) | 85.20 | 82.14 | 44.00 | 39.60 | **44.08** | **93.00** | 40.99 | 95.72 | 60.58 | **59.83** | **61.60** | **64.25** |

Table 1: Main results on zero-label performance of different methods. All methods are based on the T0-11B model. "EnsD" denotes ensemble distillation.

**Few-Shot Performance.** Next, we turn to experiments where 32 labeled samples are available for each task. Table 2 shows that MT + ZPS outperforms all other strong few-shot baselines on the average performance of all test tasks. This reveals the effectiveness of our ZPS. We notice that the performance of ICL with T0 is significantly worse than other few-shot baselines, which is probably because the priming prompt used in ICL is quite different from task descriptions used in the multitask training stage of T0. The performance of GRIPS and PT are similar, slightly better than the T0 zero-shot baseline, while GPS achieves the strongest performance among all few-shot methods that do not require parameter updating. In our MT + GPS, with the help of pseudo labels, more labeled data can be freed to perform training instead of model selection (traditional MT requires a portion of the labeled set to select checkpoints (Zheng et al., 2021)). The superior effectiveness of ZPS validates that pseudo-labeled data can not only select prompts, but select model checkpoints as well.

### 4.3 ABLATION STUDY

In this section, we perform several ablation experiments to explore the influential factors of our method. We keep the other factors fixed while studying the effect of a specific factor.

| | Natural Language Inference | | | | | Sentence Completion | | | Coreference | | WSD | Avg. |
|---|---|---|---|---|---|---|---|---|---|---|---|---|
| | RTE | CB | ANLI1 | ANLI2 | ANLI3 | COPA | Hella. | Story. | WSC | Wino. | WiC | |
| GPS | **83.86** | 80.00 | 46.47 | 39.91 | 43.01 | **93.43** | 43.96 | 95.03 | **65.48** | 61.96 | 58.82 | 64.72 |
| GRIPS | 81.59 | 76.07 | 44.41 | 39.32 | 42.10 | 91.41 | 33.11 | 94.16 | 61.54 | 58.61 | 57.23 | 61.78 |
| ICL | 72.42 | 65.24 | 37.58 | 33.05 | 37.17 | 84.07 | 26.77 | 90.18 | 64.42 | 54.21 | 49.33 | 55.86 |
| PT | 81.48 | 70.24 | 42.67 | 38.67 | 40.55 | 92.59 | 39.12 | **95.46** | 60.87 | 59.79 | **58.90** | 61.85 |
| MT | 78.34 | 81.25 | **47.10** | 38.43 | 45.75 | 91.45 | 52.91 | 94.55 | 65.38 | 69.53 | 53.29 | 65.27 |
| MT + ZPS (Ours) | 80.87 | **85.71** | 45.50 | **41.60** | **50.33** | 93.00 | **58.54** | 93.80 | 57.69 | **69.93** | 53.92 | **66.45** |

Table 2: Main results on few-shot performance of different methods. All methods are based on the T0-11B model. For a fair comparison, we keep the size of the labeled set for each task as 32. Model Tuning (MT) and Prompt Tuning (PT) tune the full model parameters and the continuous prompt embedding separately. In-Context Learning (ICL), GRIPS (Prasad et al., 2022) and GPS (Xu et al., 2022) do not require any parameter update.

### 4.3.1 ENSEMBLE STRATEGIES

As shown in Table 3, among these three ensemble strategies, log-probability mean attains the best performance. This also echoes with the finding in Jiang et al. (2020) that given any prompt, log probabilities can penalize the choice that is very unlikely.

| | Natural Language Inference | | | | | Sentence Completion | | | Coreference | | WSD | Avg. |
|---|---|---|---|---|---|---|---|---|---|---|---|---|
| | RTE | CB | ANLI1 | ANLI2 | ANLI3 | COPA | Hella. | Story. | WSC | Wino. | WiC | |
| Majority vote | 86.28 | 78.57 | 44.30 | 39.30 | 42.92 | 92.00 | 33.03 | 95.35 | 62.50 | 59.75 | 59.40 | 63.04 |
| Probability mean | 86.28 | 78.57 | 44.30 | 39.30 | 42.92 | 92.00 | 33.79 | 95.83 | 56.73 | 59.75 | 59.40 | 62.74 |
| Ours (log prob mean) | 86.28 | 80.36 | 45.10 | 40.50 | 42.92 | 92.00 | 34.33 | 95.83 | 62.50 | 60.54 | 58.78 | 63.54 |

Table 3: Performances of different ensemble strategies.

### 4.3.2 PROMPT FILTERING

Prompt filtering plays an important role in our method. We compare the performance of using and not using filtering in Table 4. It shows that prompt filtering boosts the performance of ZPS for almost all tasks. Since prompt filtering is followed by a prompt ensemble, filtering out possible low-performing prompts contributes to a higher accuracy of the ensemble and leads to better performance. Moreover, even without prompt filtering, our ZPS still holds strong advantages over the other baselines in Table 1, indicating the stability and robustness of our method.

| | Natural Language Inference | | | | | Sentence Completion | | | Coreference | | WSD | Avg. |
|---|---|---|---|---|---|---|---|---|---|---|---|---|
| | RTE | CB | ANLI1 | ANLI2 | ANLI3 | COPA | Hella. | Story. | WSC | Wino. | WiC | |
| w/o filter | 81.95 | 78.57 | 44.80 | 39.30 | 44.41 | 92.00 | 34.33 | 95.35 | 62.50 | 59.70 | 56.13 | 62.96 |
| w/ filter | 86.28 | 80.36 | 45.10 | 40.50 | 42.92 | 92.00 | 34.33 | 95.83 | 62.50 | 60.54 | 58.78 | 63.54 |

Table 4: Results of ZPS with and without filtering.

### 4.3.3 ROBUSTNESS TO ADVERSARIAL PROMPTS

Since ZPS relies on a prompt ensemble, it is possible that the quality of the candidate prompts will affect its performance. We want to test whether ZPS will still outperform the average performance of candidate prompts when there are more low-performing prompts in the candidates. To this end, we create adversarial low-performing prompts by adapting GPS (Xu et al., 2022). In the genetic algorithm, the generated prompts with the lowest performance on the validation set are selected as the low-performing prompts. A portion of the original prompts are replaced with the same number of low-performing prompts. We then report the performance of ZPS-selected prompts and the average of candidates. The replacement process is repeated with 5 different random seeds. We also vary

the ratio of low-performing prompts from 0.1 to 0.8. As shown in Table 5, ZPS yields consistent improvements over not using prompt selection (i.e., the mean of candidate prompts). This reflects that ZPS is useful consistently under different levels of adversary and noise.

| Ratio | 0.1 | 0.2 | 0.5 | 0.8 |
|---|---|---|---|---|
| ZPS | 62.20 | 62.07 | 58.98 | 51.45 |
| No prompt selection | 59.95 | 58.45 | 54.90 | 50.93 |

Table 5: Robustness to adversarial prompts. We replace the original candidate prompts with adversarial low-performing prompts generated by GPS (Xu et al., 2022). We vary the ratio of low-performing prompts from 0.1 to 0.8. ZPS yields consistent improvements over not using prompt selection.

### 4.3.4 SCALE

Another important factor is the model scale. We conduct experiments on T0-3B and T0-Large(770M) to verify the effectiveness of ZPS on different models. Table 6 shows that ZPS is able to boost the performance on models with different scales.

| | | Natural Language Inference | | | | | Sentence Completion | | | Coreference | | WSD | Avg. |
|---|---|---|---|---|---|---|---|---|---|---|---|---|---|---|
| | | RTE | CB | ANLI1 | ANLI2 | ANLI3 | COPA | Hella. | Story. | WSC | Wino. | WiC | |
| 3B | T0 | 60.61 | 48.81 | 35.11 | 33.28 | 33.52 | 73.44 | 27.76 | 84.91 | 65.00 | 50.91 | 51.27 | 51.32 |
| | ZPS (ours) | 58.48 | 60.71 | 36.60 | 34.40 | 33.33 | 76.00 | 28.49 | 87.39 | 64.42 | 51.78 | 50.63 | 52.93 |
| Large | T0 | 72.67 | 56.55 | 32.77 | 32.15 | 34.38 | 85.36 | 27.18 | 93.04 | 63.94 | 54.35 | 50.33 | 54.79 |
| | ZPS (ours) | 79.06 | 67.86 | 31.20 | 31.10 | 34.25 | 88.00 | 29.16 | 93.43 | 65.38 | 53.43 | 49.84 | 56.61 |

Table 6: The performance of ZPS with different scales.

### 4.3.5 USAGE OF LABELED AND PSEUDO-LABELED DATA

In this part, we examine different ways of using labeled and pseudo-labeled data in the few-shot setting. Given 32 labeled examples $\mathcal{D}_{32}$ from the training set and all unlabeled examples $\mathcal{X}$ for each task, we consider the following cases:

- 16 + 16: This is a classic approach to use labeled data in model tuning. The labeled set is splited into two equal-sized halves, $\mathcal{D}_{16}$ and $\mathcal{D}'_{16}$, as a training set and a validation set. The validation set is then used to perform checkpoint selection and prompt selection.

| | | Natural Language Inference | | | | | Sentence Completion | | | Coreference | | WSD | Avg. |
|---|---|---|---|---|---|---|---|---|---|---|---|---|---|---|
| | | RTE | CB | ANLI1 | ANLI2 | ANLI3 | COPA | Hella. | Story. | WSC | Wino. | WiC | |
| 16 + 16 | Mean | 78.19 | 72.38 | 49.16 | 38.87 | 45.94 | 90.45 | 46.36 | 94.55 | 63.65 | 70.04 | 53.18 | 63.88 |
| | Val select | 78.34 | 81.25 | 47.10 | 38.43 | 45.75 | 91.45 | 52.91 | 94.55 | 65.38 | 69.53 | 53.29 | 65.27 |
| | Pseudo select | 80.51 | 73.21 | 48.20 | 39.50 | 46.25 | 91.00 | 52.91 | 94.23 | 64.42 | 70.09 | 52.51 | 64.80 |
| 32 pseudo train | Mean | 81.91 | 79.88 | 38.89 | 36.20 | 42.13 | 93.65 | 51.94 | 94.12 | 56.54 | 57.47 | 55.11 | 62.53 |
| | Val select | 82.43 | 79.76 | 38.80 | 36.50 | 42.30 | 96.00 | 51.51 | 94.07 | 61.54 | 57.73 | 55.33 | 63.27 |
| | Pseudo select | 81.95 | 82.14 | 39.00 | 36.50 | 43.17 | 95.00 | 51.51 | 94.33 | 57.69 | 57.46 | 55.33 | 63.10 |
| 32 pseudo val | Mean | 83.72 | 80.60 | 48.97 | 37.98 | 46.01 | 92.55 | 48.06 | 95.79 | 65.48 | 70.54 | 55.38 | 66.00 |
| | Pseudo select | 83.75 | 78.57 | 49.40 | 37.50 | 45.58 | 93.00 | 56.10 | 96.26 | 62.50 | 70.40 | 55.96 | 66.28 |
| More pseudo val | Mean | 79.64 | 85.00 | 46.01 | 40.74 | 50.16 | 92.80 | 51.62 | 93.67 | 62.88 | 70.12 | 56.54 | 66.32 |
| | Pseudo select | 80.87 | 85.71 | 45.50 | 41.60 | 50.33 | 93.00 | 58.54 | 93.80 | 57.69 | 69.93 | 53.92 | 66.45 |

Table 7: An ablation study on different strategies to use labeled and pseudo-labeled data. "Mean" denotes the mean of prompts, "val select" means selecting the prompt on the labeled validation set, and "pseudo select" means prompt selection on the pseudo-labeled data. We come to the conclusion that it is best to use labeled data for training and use pseudo-labeled data for checkpoint and prompt selection. "More pseudo val" with "pseudo select" is what we adopt in ZPS.

- 32 pseudo train: We first annotate the unlabeled test data $\mathcal{X}$ by T0 to get pseudo labels. Then, we select the pseudo-labeled samples with the top-32 highest confidence. This set is denoted as $\mathcal{X}_{32}$. The entire labeled set $\mathcal{D}_{32}$ is only used as a validation set.

- 32 pseudo val: This case is similar to "32 pseudo train", despite that the entire labeled set $\mathcal{D}_{32}$ is used as a training set for model tuning. In this case, we use $\mathcal{X}_{32}$ for checkpoint selection.

- More pseudo val (the approach we adopted in our ZPS): The only difference between this case and "32 pseudo val" is that we increase the size of the pseudo-labeled validation set.

Table 7 presents the results of the aforementioned cases. These four cases all show significant improvements compared with the T0 zero-label baseline, which demonstrated the effectiveness of pseudo-labeling. "32 pseudo train" performs the worst among all cases, indicating that the pseudo-labeled samples still contain much noise that is harmful for training. By comparing "16 + 16" and "32 pseudo val", we show that the prompt selecting ability of ZPS is comparable to that of a labeled validation set. Moreover, "Pseudo val" methods perform much better than others, which validates our hypothesis. These results indicate that the gain of training with the labeled set is greater than using the labeled set for checkpoint selection. However, since the pseudo-labeled data is noisy, increasing the size of pseudo-labeled validation data can only provide marginal performance gain. Overall, we adopt "more pseudo val" with "pseudo select" in our ZPS for the best performance.

## 4.4 CASE STUDY

| Prompt | $\mathcal{D}_{16}$ Acc. | Pseudo Acc. | Acc. |
|---|---|---|---|
| Given {{premise}} Is it guaranteed true that "{{hypothesis}}"? Yes or no? | 87.88 | 92.06 | 81.95 |
| Suppose {{premise}} Can we infer that "{{hypothesis}}"? Yes or no? | 87.88 | 91.34 | 81.95 |
| {{premise}}\n Question: {{hypothesis}} True or False? | 84.85 | 94.58 | 82.31 |
| {{premise}}\n\n Question: Does this imply that "{{hypothesis}}"? Yes or no? | 90.91 | 90.97 | 82.31 |
| Given {{premise}} Should we assume that "{{hypothesis}}" is true? Yes or no? | 90.91 | 94.22 | 81.95 |
| Given that {{premise }} Does it follow that {{hypothesis}} Yes or no? | 90.91 | 81.95 | 73.29 |
| {{premise}} Based on the previous passage, is it true that "{{hypothesis}}"? Yes or no? | 90.91 | 94.95 | 86.28 |
| {{premise}} Are we justified in saying that "{{hypothesis}}"? Yes or no? | 84.85 | 86.64 | 75.81 |
| Given that {{premise}} Therefore, it must be true that "{{hypothesis}}"? Yes or no? | 90.91 | 94.58 | 82.31 |

Table 8: The performance of RTE prompts illustrated by: **(1)** $\mathcal{D}_{16}$ Acc., the accuracy of a validation set with 16 labels **(2)** Pseudo Acc., the pseudo accuracy obtained by ZPS, and **(3)** Posthoc Acc., the true accuracy on the test set.

In Table 8, we present 10 RTE prompts and the following quantities **(1)** $\mathcal{D}_{16}$ Acc., which is the accuracy of a prompt on a small validation set with 16 labeled samples. **(2)** Pseudo Acc., which is calculated by ZPS and **(3)** Posthoc Acc., which is the real performance on the test set. The results show that $\mathcal{D}_{16}$ has poor prompt selection abilities and there are no obvious correlations between $\mathcal{D}_{16}$ Acc. and Posthoc Acc. On the contrary, our ZPS can distinguish the difference in prompt performances and have better correlations with the real test performance.

## 5 CONCLUSIONS

We propose a novel Zero-Label Prompt Selection (ZPS) algorithm to select a high-performing prompt in the challenging setting where no labeled data is available. We demonstrate that ZPS can be used in a plug-and-play manner for different cases (zero-label generalization, ensemble distillation, and few-shot model tuning). We also perform detailed ablation studies to verify the robustness and effectiveness with respect to different factors.

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
