# OpenReview forum: "Zero-Label Prompt Selection"
_ICLR.cc/2023/Conference — Submitted to ICLR 2023_

### Official Review · Reviewer_34nz · 2022-10-17

**Confidence:** 4
**Correctness:** 3
**Technical Novelty And Significance:** 2
**Empirical Novelty And Significance:** 3
**Recommendation:** 5

**Clarity, Quality, Novelty And Reproducibility:**

The paper is clearly written.
The paper is technically sound and people shall be able to replicate the results without too much difficulty, but an open-sourced repo would be favored.

**Strength And Weaknesses:**

Strength:

1. The problem is well-motivated. How to select prompts in the zero-shot setting is a very interesting and important research problem.
2. The solution is simple, yet effective.
3. The evaluation is comprehensive and the analysis is interesting.

Weakness:

1. An important and highly-related baseline is missing.
Despite being a very interesting research problem, there are actually some existing works that try to address this problem. For instance, [1]  choose the prompt that maximizes the mutual information between the input and the corresponding model prediction. Their method has shown surprising performance in prompt selection. As another example, Lu et al., 2022 also try to construct a pseudo validation set for prompt selection.  Although their method of creating the validation set is quite different from ZPS, the intuition behind is very similar and worth further discussion in the related work. The existence of [1] hurts the novelty of ZPS, and also brings an important baseline to compare with.


[1] An Information-theoretic Approach to Prompt Engineering Without Ground Truth Labels.


**Summary Of The Paper:**

This paper proposes Zero-Label Prompt Selection (ZPS), an algorithm to select the best prompt without any labeled data. In practice, they first use a heuristic rule to filter out low-quality prompts and obtain a candidate prompt set. Given these candidate prompts, they use them to assign pseudo labels for some unlabeled X in the dataset, using different ensemble strategies. This will result in a "probing set", which can be used as a pseudo validation set to select the best prompt. They also extend ZPS to support few-shot settings.

**Summary Of The Review:**

This paper targets an interesting problem and presents a simple yet effective solution. However, highly-similar work already exists (but missing) which hurts this submission's novelty. This missing related work has to be discussed and even compared to demonstrate the effectiveness of ZPS.

---

### Official Review · Reviewer_g5DF · 2022-10-23

**Confidence:** 4
**Correctness:** 3
**Technical Novelty And Significance:** 2
**Empirical Novelty And Significance:** 2
**Recommendation:** 3

**Clarity, Quality, Novelty And Reproducibility:**

This paper is well-written and very clear about the contributions of the proposed method. Also, the method is simple and easy to reproduce. However, the method and the result is not novel enough considering that there is a lot of related work on improving the performance of LMs in zero-shot or few-shot setting. Also, the evaluation setting has some weaknesses to prove the effectiveness of the proposed method.

**Strength And Weaknesses:**

Strengths:
1. The proposed method is simple and straightforward. It is easy to reproduce the proposed method.
2. The effectiveness of the proposed method is shown in both zero-shot and few-shot settings.
3. The paper is well-written and easy to understand.


Weakness:
1. It seems like the whole unlabeled training instances are used for ZPS (zero-shot). This cannot be seen as a realistic setting. I wonder if ZPS for zero-shot setting also shows effectiveness when there are not many unlabeled training instances (~16).
2. The paper claims that they do not serve prompt ensemble during inference and instead select a single prompt to use the selected prompt for inference due to computational cost or latency. However, applying ZPS also increases the computational cost and latency if it uses many unlabeled training instances to select a single prompt (sometimes even more than prompt ensembling during inference when the training instances are larger than evaluation instances). Therefore, the performance of ZPS should also be compared against prompt ensembling during inference for a fair comparison.
3. Is ZPS applicable for other prompt libraries (different from promptsource) or other model families (GPT-3, OPT, etc)? I think that showing that ZPS works for LMs that are not fine-tuned to follow instructions is important because the prompt sensitivity is known to be much higher and problematic for general pretrained LMs.
4. The performance of zero-label prompt selection is highly dependent on the quality of prompt candidates. If there are not any "good" prompts that are manually created by humans, the upper bound of the performance would be low. The implicit assumption made by the paper, "there is at least 1 good prompt", is not a realistic assumption for the real-world use case. Also, ZPS cannot be applied to benchmarks that only consist of a single prompt such as BIG-bench because ZPS does not generate but select an optimal prompt instead. It would underperform GRIPS baseline which "edits" the instruction when all of the candidate prompts show low performance without edit operation.
5. The paper does not compare ZPS with the oracle performance (the performance when selecting the best-performing prompt). The performance should also be compared with oracle performance and also the accuracy of how often the oracle prompt is selected. This accuracy directly shows the performance of "prompt selection".
6. For few-shot ZPS, MT+ZPS uses twice more training instances (32) compared to baselines and uses unlabeled data for checkpoint and prompt selection. However, although I agree that we can use more labeled data for training by applying ZPS on unlabeled data for validation, we cannot accurately see MT+ZPS is more effective than MT itself; two factors, which are 1) effect of using more labeled data for training and 2) effect of applying ZPS, are not separated. Therefore, it is hard to claim that ZPS is effective on few-shot setting by only this result. There should be baseline results on 32 examples as well to see 2).
7. Although ZPS is effective for 11B scale, for 3B scale, it underperforms naive T0 on 4 out of 11 datasets which is free from any prompt ensemble techniques and the need for unlabeled data. This shows that the effectiveness of ZPS is not universal.

**Summary Of The Paper:**

This paper introduces zero-label prompt selection (ZPS), which selects the optimal prompt for an unseen task without any labeled data or gradient update. ZPS obtains pseudo labels of an unseen task through prompt ensemble, calculates pseudo accuracy by comparing pseudo labels with the prediction of each prompt, and selects the prompt candidate with the highest pseudo accuracy. This paper shows the effectiveness of ZPS in both zero-shot and few-shot training settings.

**Summary Of The Review:**

This paper proposed a simple but effective method to select the optimal prompt for an unseen task. The proposed method, Zero-label prompt selection (ZPS) boosts the performance of the T0 model and outperforms baselines such as self-training for zero-shot setting and model tuning and GRIPS method for few-shot setting. Although the paper is straightforward and well-written, the evaluation setting has some weaknesses. First, the assumption that a lot of unlabeled data for the target task is unrealistic. Also, the experiments of this paper are limited to the T0 model and promptsource library although the method could be applied to other types of model: more extensive experiments are needed to show the effectiveness of the proposed model. ZPS also has a practical limitation that the performance is dependent on the prompt candidate; if there are no "good" prompts, the effectiveness would be small. Lastly, for few-shot setting experiment, the baselines should be trained with the same number of labeled instances to show that ZPS is effective for few-shot setting as well.

---

### Official Review · Reviewer_xSND · 2022-10-24

**Confidence:** 4
**Correctness:** 3
**Technical Novelty And Significance:** 2
**Empirical Novelty And Significance:** 2
**Recommendation:** 5

**Clarity, Quality, Novelty And Reproducibility:**

The paper is very clear and easy to be understood. Though the proposed algorithm outperforms prior methods in some downstream tasks, it is more like a trick so the novelty is limited. The code of the paper has not yet released, but it is not difficult to be reproduced through its methodology.

(1-4 by order)

Clarity: 4

Quality: 3

Novelty: 2

Reproducibility: 3

**Strength And Weaknesses:**

Strength:
1. The paper is well organized to be understood clearly.
2. A large number of experiments are conducted on downstream tasks and the results show that the introduction of ZPS can boost the performances of baseline methods.

Weakness:
1. In the part of prompt filtering, the confidence score and the role of clustering are not fully explained from the mathematical perspective. The relationship between the confidence score and the performance of prompts is not clear.
2. The proposed prompt selection algorithm is too empirical and has limited novelty.

**Summary Of The Paper:**

The paper proposes a zero-label prompt selection (ZPS) algorithm and extend it to few-shot tasks. The algorithm can help select high-performing prompts and be used in both zero-label and few-shot settings conveniently.

**Summary Of The Review:**

The core idea of the paper focuses on prompt selection. Yet, it lacks valid explanation and experiments to show why the use of the confidence score can filter low-performing prompts. Although the experiments show that the introduction of this algorithm improves the result of baselines, it lacks comparison with same type of algorithms (i.e., prompt selection algorithms).

---

### Official Review · Reviewer_5fB6 · 2022-10-24

**Confidence:** 5
**Correctness:** 3
**Technical Novelty And Significance:** 2
**Empirical Novelty And Significance:** 4
**Recommendation:** 6

**Clarity, Quality, Novelty And Reproducibility:**

This paper is clear but it's not technically very novel. I urge the authors to release the code if accepted.

**Strength And Weaknesses:**

## Strengths

1. This paper explores pseudo-labeling at scale.
1. This paper is well-written and easy to follow.
1. The performance is good compared to self-training and the original T0.
1. This paper sheds light on how to evaluate the effectiveness of a prompt, which previously highly relies on human expertise.
1. This paper also explores robustness and ensemble strategies. I found the analysis interesting and informative.

## Weaknesses

1. The technical novelty is somewhat limited. The technique of determining pseudo-accuracy is not new.
1. This paper is built on an assumption that a small validation set is not available. However, I would like to see how it's compared to select prompts on a very small validation set (e.g., with only 20 or 50 examples annotated by humans), which may be a more realistic setting.

**Summary Of The Paper:**

This paper proposes Zero-Label Prompt Selection (ZPS) that selects prompts without any labeled data or gradient update.

**Summary Of The Review:**

Despite concerns about the novelty, I recommend this paper because of its potential impact on prompt learning.

---

### Decision · Program_Chairs · 2023-01-20

**Decision:**

Reject

**Justification For Why Not Higher Score:**

The authors did not respond to some crucial questions raised by the reviewers.

**Justification For Why Not Lower Score:**

N/A

**Metareview: Summary, Strengths And Weaknesses:**

Summary: The paper is tackling the problem of prompt selection when no label is available. While the proposed method shows non-trivial improvement over the baselines, many questions were raised by the reviewers, including whether such setting (assumption) is realistic, but the authors did not respond to these questions in the rebuttal period.

Strengths:
- Most reviewers agree that the proposed method is simple and effective.

Weaknesses:
- Most reviewers think that the novelty of the paper is limited.
- Most reviewers think that zero-label is not a realistic setting.
- Most reviewers have raised many questions about unclear parts in the paper, but they were not addressed by the authors during the rebuttal period.